# Complication Prediction after Esophagectomy with Machine Learning

**DOI:** 10.3390/diagnostics14040439

**Published:** 2024-02-17

**Authors:** Jorn-Jan van de Beld, David Crull, Julia Mikhal, Jeroen Geerdink, Anouk Veldhuis, Mannes Poel, Ewout A. Kouwenhoven

**Affiliations:** 1Faculty of EEMCS, University of Twente, 7500 AE Enschede, The Netherlands; 2Hospital Group Twente (ZGT), 7609 PP Almelo, The Netherlands; 3Faculty of BMS, University of Twente, 7500 AE Enschede, The Netherlands

**Keywords:** esophagectomy, clinical decision support, multimodal machine learning, temporal learning

## Abstract

Esophageal cancer can be treated effectively with esophagectomy; however, the postoperative complication rate is high. In this paper, we study to what extent machine learning methods can predict anastomotic leakage and pneumonia up to two days in advance. We use a dataset with 417 patients who underwent esophagectomy between 2011 and 2021. The dataset contains multimodal temporal information, specifically, laboratory results, vital signs, thorax images, and preoperative patient characteristics. The best models scored mean test set AUROCs of 0.87 and 0.82 for leakage 1 and 2 days ahead, respectively. For pneumonia, this was 0.74 and 0.61 for 1 and 2 days ahead, respectively. We conclude that machine learning models can effectively predict anastomotic leakage and pneumonia after esophagectomy.

## 1. Introduction

According to the World Cancer statistics in 2020, esophageal cancer ranks seventh in terms of incidence and sixth in mortality overall. The disease is more common in men (70%) and most prevalent in eastern Asia [1].

In the past decade, minimally invasive robot-assisted techniques have become increasingly popular as an alternative to open esophagectomy. Clinical trials have shown that minimally invasive procedures lower the risk of postoperative complications, specifically, pulmonary complications [2,3]. Nevertheless, postoperative complications are common, with a rate of 65% reported by a nationwide study in the Netherlands including 1617 patients [4]. In this study, pneumonia (21%) and anastomotic leakage (AL) (19%) were the most common postoperative complications. A similar study with 2704 patients from centers across 14 different countries found a postoperative complication rate of 59% and rates of 14.6% and 11.4% for pneumonia and AL, respectively [5].

In recent years, there have been considerable advances in the field of medical artificial intelligence (MAI) with successes in a wide range of retrospective studies [6]. For example, pneumonia detection models have been developed, with most reporting an accuracy over 90%; logistic regression (LR) and deep learning (DL) models are most commonly used for this task [7]. Yet, great challenges remain in the field of MAI, for example, the development of multimodal models that can handle various medical data sources as input [6].

Machine learning methods have been employed to analyze and predict complications post-esophagectomy. Early studies, often limited to preoperative variables, focused on the identification of preoperative risk factors, but agreement about these factors is limited [8,9,10]. A recent review showed that the use of deep learning methods is limited [11], still, AUROC scores of up to 0.96 have been reported for the diagnosis of AL [12,13]. Crucially, these studies included postoperative laboratory results: C-reactive protein (CRP), lymphocyte count, and albumin. A post-esophagectomy complication prognosis model could improve postoperative decision-making, speed up recovery, and reduce hospital stays [14].

In this paper, we study to what extent machine learning methods can predict and detect complications after esophagectomy. Specifically, we consider the two most common infectious complications: pneumonia and AL. Additionally, we study to what extent the temporality of the data is relevant for early detection, where temporality refers to the day-to-day measurement of clinical variables. Our dataset spans four modalities: preoperative patient characteristics, laboratory results, vital signs, and radiology images. We compare models ranging in complexity and the number of considered modalities, where we start with logistic regression (LR) and end with a multimodal temporal deep learning model. Our models achieve state-of-the-art performance in the prediction and detection of pneumonia and AL.

The remainder of this paper is structured as follows: in Section 2, we discuss the adopted materials and methods. Section 3 presents the results, followed by a discussion in Section 4. We present conclusions relevant to the work in Section 5.

## 2. Materials and Methods

The data used in this work are obtained from the Electronic Health Records database of Hospital Group Twente (in Dutch Ziekenhuisgroep Twente (ZGT)) in the Netherlands. Below is a comprehensive description of this dataset, followed by the data preparation steps. The end of this section discusses our machine learning models and training methodology.

### 2.1. Data Description

The dataset contains 417 patients who underwent esophagectomy between 2011 and 2021 at ZGT. Patients either underwent the Ivor Lewis (*n* = 326) or McKeown (*n* = 88) procedure. We collected data after surgery up until discharge to a maximum of 14 days from four distinct modalities: preoperative patient characteristics, laboratory results, vital signs, and radiology images. The dataset was annotated by a clinician for complications, including the date of complication.

#### 2.1.1. Complications

The dataset specifies two types of complications, anastomotic leakage (AL) and pneumonia, where the latter was determined according to the Utrecht Pneumonia Score (UPS) [15].

Figure 1a shows how these complications are distributed over the days after surgery, with day 0 as the day of surgery. The data confirm complications are common after esophagectomy with a rate of 32% for pneumonia and 15% for AL. Furthermore, there is a clear difference in distribution between the two complication types. Pneumonia is most prevalent during the first 5 days after surgery, while AL tends to be diagnosed between days 5 and 10 after surgery.

It is possible for complications to co-occur, meaning some patients were diagnosed with both complications at some point after surgery. This is the case for 30 patients, out of whom 24 were diagnosed with AL shortly after being diagnosed with pneumonia, in the other cases the order of complications was reversed or coincided.

Figure 1b compares the length of stay between patients with and without postoperative complications. Patients with postoperative complications tend to stay longer, with 51% of them staying for more than 14 days. Patients without postoperative complications are usually discharged between days 6 and 10 after esophagectomy.

#### 2.1.2. Preoperative Characteristics

We collected information known before surgery from the 417 patients in our dataset, statistics of these variables are shown in Table 1. We can estimate their predictive power for leakage and pneumonia with statistical significance tests. We applied the Student’s *t*-test and proportions z-test for continuous and binary values, respectively. The assumptions for these tests were verified. Only Weight had a *p*-value smaller than 0.025 for pneumonia, suggesting our selection of preoperative patient characteristics holds little predictive power for postoperative complications.

#### 2.1.3. Laboratory Results

Three types of laboratory measurements were taken daily after esophagectomy: C-reactive protein (CRP), leukocytes, and amylase. CRP and leukocytes are blood tests and are known as indicators for infection [16]. Amylase is determined from drain fluid and an indicator specific to AL [17]. Other laboratory measurements have been excluded based on data availability.

In Figure 2, we compare amylase and CRP values with respect to AL and pneumonia. First, we observe that amylase values tend to be higher for patients with AL, albeit with multiple outliers in the patient group without AL. Second, a similar relation appears between CRP and pneumonia, but with more overlap between the groups. Furthermore, a clear postoperative pattern is visible, where CRP values first rise, due to surgery, and then slowly decline until day 8. Boxplots for all combinations of laboratory results and complications can be found in Appendix A.

#### 2.1.4. Vital Signs

After surgery, vital signs are regularly measured to monitor the patient, specifically, heart rate, systolic blood pressure, temperature, and respiratory rate. These are automatically measured while patients are in the intensive care unit (ICU), otherwise, they are taken manually. Each day is split into three 8 h windows (00:00–07:59, 08:00–15:59, 16:00–23:59) from which we have the maximum value, except for systolic blood pressure where we have the minimum value.

The raw dataset contained some unrealistic values for the vital signs, and together with a clinician, we set an acceptable range for each vital sign. Table 2 shows these ranges, any values outside these ranges are treated as missing.

Temperature is a part of the UPS; in Figure 2 we show the boxplot with respect to pneumonia. Temperature seems to be slightly elevated for patients with pneumonia. Still, there is a strong overlap between patient groups. The boxplots for the other vital signs did not show a strong correlation as may be seen in Appendix A.

#### 2.1.5. Radiology Images

Various imaging techniques are used to track recovery and detect postoperative complications. Moreover, infiltrative findings on chest X-ray images are required to assign pneumonia according to the UPS [15]. After esophagectomy, chest X-ray images are taken on close to a daily basis during the first week after surgery, where the frequency of scans depends on patient recovery.

All chest X-ray images taken within 14 days after surgery were collected. We exclusively consider images taken in the anterior–posterior (AP) view or posterior–anterior (PA) view. AP view images are commonly taken while the patient is lying in bed, while PA view images are taken from a standing position.

### 2.2. Data Preparation

The raw dataset required further preparation before training our machine-learning models. There are two prediction windows: 1 (24 h) and 2 (48 h) day(s) ahead of a complication. Additionally, we include a detection window corresponding to the day of the complication (0 h). The preprocessing steps, repeated before each cross-validation run, are summarized below:Sequence trimming;Imputation missing data;Subsampling sequence length majority class.

In the first step, we trim data based on the target complication, length of stay, and prediction window. Data after the prediction window is excluded for patients with a complication. For example, if a patient was diagnosed with pneumonia on day 5 and we want to predict on day 4, then we only include data up until day 4 and exclude subsequent data. Furthermore, we exclude patients for whom the given complication was diagnosed before day 4 after surgery to ensure multiple days of data for our temporal models.

Next, we use forward filling to impute missing sequential data. This filling strategy does not consider future data, thereby emulating real-time clinical decision making.

Last, we trim the sequences of patients without complications (negative class), such that the sequence length distribution equals that of patients with a complication (positive class). Figure 3 shows the distributions before and after the subsampling operation for AL. For example, without subsampling the models would likely predict no leakage on day 15 regardless of feature values.

### 2.3. Predictive Models

We employed a range of machine learning models with various levels of complexity and numbers of considered modalities. We started with logistic regression, which has been used for postoperative complication prediction after esophagectomy [18]. This study used demographics and preoperative serum values to predict pulmonary complications with an AUROC of 0.71.

Once we established our performance baseline with logistic regression we shifted to more complex models, beginning with a model that can make a prediction using images. Next, we trained a temporal model that takes a sequence of daily features as input. Initially, we applied this model to each modality separately, so we ended up with a set of pre-trained unimodal models. Afterward, we fused these unimodal models to form multimodal models that can take all modalities as input simultaneously. All models are described in more detail below.

#### 2.3.1. Logistic Regression (LR)

For our baseline model, we used LR which took data from a single day as input to predict postoperative complications at our three time windows. At this stage we excluded images as they are incompatible with LR, leaving us with three modalities: laboratory results, vital signs, and preoperative characteristics. The vital signs were processed further by taking the daily minimum or maximum appropriately; see Section 2.1. In total, we trained five LR models, one for each modality (*n* = 3) and two multimodal LR models, taking up to three modalities as input.

We used the logistic regression classifier (https://scikit-learn.org, accessed on 8 August 2023). All settings were left to default, except for the inverse regularization strength C, which was optimized empirically by taking a value between $10^{-3}$ and $10^{3}$.

#### 2.3.2. Imaging

AI algorithms have been successfully trained for various radiological tasks ranging from detection and localization of abnormalities to radiology worklist ordering [19]. Convolutional neural networks (CNNs) are commonly used for image classification. Specifically, for our task, we used the ResNet18 architecture [20]. We elected this relatively small network with less trainable parameters because of our limited training dataset.

The images were split into two groups depending on whether the corresponding patient had a postoperative complication. Furthermore, we only considered the latest image at a given window before complication for the positive group. This results in severe class imbalance due to the limited number of positively labeled images. Therefore, during training, we increased the weight of positive images.

#### 2.3.3. Temporal Models

Next, we considered data taken over more than one day as input to our models, where the number of days ranged from 3 to 14, as described in Section 2.2. We used long short-term memory (LSTM) units to extract features from our sequential data [21]. The temporal lab and vitals models are similar but have a different input frequency of once and thrice a day, respectively. For the sequence of images, we first extracted features from each image separately with a pretrained ResNet18, then we fed these features into an LSTM layer.

#### 2.3.4. Multimodal Deep Learning Models

In a multimodal model, information from disparate data sources is combined to improve robustness and performance. Commonly, a model for each data source is trained in isolation, then these unimodal models are fused to form a single multimodal model. Multiple fusion strategies have been used in medical machine learning and come in three broad categories: early, intermediate, and late fusion [22].

The most straightforward method is late fusion, where unimodal predictions are concatenated and presented to a classifier, essentially leading to a weighted average. Late fusion is comparatively easy to implement; however, it prohibits interaction between data sources [22]. For example, in our dataset a late-fusion model would not be able to directly link an infiltrate in the thorax image with an increased temperature, yet both are part of the UPS we have used to diagnose pneumonia.

During early fusion, data sources are first mapped to a single feature vector before entering a classifier [23]. This fusion method encourages information flow between data sources and does not require unimodal classifiers. Typically, the mapping method is static and not improved upon during training.

Lastly, intermediate, mid-, or joint fusion occurs between early and late fusion [22]. First, a feature vector is extracted from each data source instead of a prediction like during late fusion, then these feature vectors are combined and given to a classifier. Mid-fusion acts similar to early fusion, but dynamically learns the mapping of disparate data sources during training.

In this study, we compared two fusion methods; late and mid-fusion. In both cases, a unimodal model was trained separately for each of the data sources. Figure 4 illustrates how the unimodal models were fused. During late fusion, the individual predictions were taken and during mid-fusion, the outputs of the pre-prediction layers (LSTM output) were concatenated. Thereby, our final multimodal models received all available data from all modalities.

### 2.4. Model Training

Models were tasked to either predict leakage or pneumonia at three prediction windows: 0 (0 h), 1 (24 h), and 2 (48 h) days ahead. We utilized repeated four-fold cross-validation since we had a small medical dataset, which can cause a high performance variance depending on the data split. All non-LR models were implemented in *Keras* and trained to minimize weighted binary cross-entropy loss, where the weights were computed with the compute_class_weight function from the scikit-learn package (https://scikit-learn.org, accessed on 25 September 2023).

We used the ROC-AUROC as the main metric to gauge the performance of our models, because of its widespread use in the literature and its insensitivity to class imbalance. Furthermore, we computed the precision, sensitivity, specificity, and F1-score, where the threshold was set such that it maximized the geometric mean of specificity and sensitivity.

Each model comes with a set of hyperparameters: learning rate, layer size, number of hidden layers, and dropout. We selected the values through Bayesian optimization that yielded the highest average AUROC. The final set of hyperparameters for each model can be found in Appendix B.

Multimodal models were trained in two steps. In the first step, we freeze the weights of the pre-trained unimodal models and only train the postfusion layers. Afterward, we unfreeze the unimodal weights and train again. Furthermore, we experiment with submodel-specific optimizers with different initial learning rates, because each model learns at its own pace. The learning rates for fine-tuning are defined as a fraction ($w_{ft}$) of the learning rate during pretraining.

Hyperparameter values and further details of each model can be found in Appendix B.

## 3. Results

The experimental results are presented in a similar order as the Section 2. Initially, we only report the AUROC for readability and quick comparison of model performance. Subsequently, we report all other metrics for selected models.

### 3.1. Logistic Regression

Table 3 shows the mean AUROC, taken over 10 × 4 cross-validation runs, for the prediction of AL and pneumonia at the three prediction windows. The static modality provides the same prediction for all windows. Therefore, we only list it at the 0 h window.

Three of our LR models achieved an AUROC of 0.92 for the prediction of AL on the same day it was diagnosed. Notably, all of these models contain the laboratory modality. Moreover, the model with just laboratory results scored best (AUROC = 0.83) for 1-day-ahead AL prediction with LR.

The model that combines laboratory results and vital signs had the highest performance for predicting pneumonia on the day of diagnosis. In particular, this modality combination improved 1-day-ahead pneumonia prediction (AUROC = 0.71), when compared to its unimodal counterparts for 1-day-ahead prediction.

Typically, models trained with the same input are better at predicting AL than pneumonia. For both prediction targets, the static preoperative patient characteristics seem to hold no predictive power at all.

### 3.2. Unimodal Models

Table 3 shows the mean AUROC of the unimodal models, which all received temporal data except for the static and single-image models. Reported values are the mean AUROC of three four-fold cross-validation runs.

The model that takes a sequence of laboratory results scored the highest test AUROC at all prediction windows for both AL and pneumonia. Furthermore, this model outperformed the LR model which takes a single day of laboratory results as input. Especially, its 0 h pneumonia prediction performance is much higher (AUROC = 0.85) than the LR counterpart (AUROC = 0.72). In contrast, the sequential vitals model does not consistently outperform the corresponding LR model.

Comparing the unimodal image models we observe that a sequence of images as input consistently outperforms a single-image model. Both image models retained similar performance across prediction windows for both targets. Furthermore, the single-image model scored better on AL prediction 1 day ahead than on the day of complication.

### 3.3. Multimodal Models

Table 3 also shows the mean AUROC of the multimodal models, which differ in fusion method and whether they were fine-tuned. On average late-fusion models consistently outperformed mid-fusion with and without fine-tuning. Generally, the multimodal models had a higher train set AUROC than their unimodal counterparts. However, only for 1-day-ahead prediction of pneumonia the test set AUROC increased by more than 1% compared to the best unimodal model. Also, joint multimodal learning (fine-tuning) did not improve the test set AUROC by more than 1%.

Table 4 lists additional metrics for the 1-day-ahead prediction of AL and pneumonia; these values are the test set averages. The LR and temporal models differed mainly in their precision scores with a difference of up to almost 40%, while differences in recall (sensitivity) and specificity remained small. However, due to our experimental design, the class imbalance for LR was larger than in the other models. Specifically, for AL the ratio for LR was 1:67 and for DL models 1:7.

## 4. Discussion

LR models predicted AL 1 day in advance with an average AUROC of 0.83; pneumonia remained more difficult to predict, with an average AUROC of 0.66 1 day in advance. These models depended only on a single day of patient data, while the temporal models had access to at least 3 days of data. The temporal models outperformed their LR counterparts, implying that the inclusion of historic measurements improves performance. Specifically, the temporal laboratory model improved the average AUROC for 1-day-ahead prediction of AL from 0.83 to 0.87. Furthermore, our best models detected (0 h) AL and pneumonia with AUROCs of 0.93 and 0.85, respectively. These diagnosis models can support clinical decision making, although prognosis is of course preferred.

In 2018, Aiolfi et al. systematically reviewed the use of CRP as an early AL predictor after esophagectomy [16]. They estimated the pooled AUROCs for AL prediction on days 3 and 5 after surgery, which were 0.80 and 0.83, respectively. Recently, a nomograph model was developed to detect AL on the day it was diagnosed. The authors reported an AUROC of 0.96 using a dataset similar in size (*n* = 308) and AL incidence rate (9.7%) [13]. Another study with 450 patients reported an AUROC of 0.95, furthermore the authors concluded that the CRP to albumin ratio is crucial for the diagnosis of AL. Therefore, the inclusion of albumin in our study could enhance the performance of our models. These AUROC scores are in line with our best-performing models, which scored an AUROC of 0.93 on the AL diagnosis task.

Van Kooten et al. studied to what extent conventional regression methods could predict post-esophagectomy pneumonia [24]. They used a dataset with preoperative characteristics from 4228 patients who underwent esophagectomy in the Netherlands between 2011 and 2017 and reported an AUROC of 0.64 with a generalized linear model. In another study, Jin et al. developed a nomograph model to predict pneumonia after esophagectomy. On their validation set, they reported an AUROC of 0.69 [25]. The models in this study using preoperative patient characteristics scored poorly (AUROC < 0.60). However, models that included temporal laboratory results predicted pneumonia well, with AUROCs of 0.85 and 0.74 on 0- and 1-day-ahead predictions, respectively.

In this study, the AUROC is the main score to evaluate the performance of our models, because it does not require a predetermined cut-off point. A value of 0.5 means a model is as good as random guessing, while a value of 1.0 indicates perfect separation of classes. Our goal is to develop a model that can support clinical decision making in our hospital, for which we defined a minimum AUROC of 0.8. Models with an AUROC below 0.6 are regarded as insufficient, while those with an AUROC between 0.6 and 0.8 are considered potentially relevant, especially in combination with other models and modalities.

The LR and DL models trained only on preoperative patient characteristics scored insufficiently, with mean AUROC test scores below 0.55. We used a limited set of preoperative variables, yet other studies did identify age and ASA-score as potential risk factors for anastomotic leakage [9,12]. Furthermore, these studies used a much broader set of variables, for example, tumor size and stage. Therefore, the predictive value of preoperative patient characteristics cannot be fully assessed within our study, because not enough preoperative values were included.

The size of our dataset is one of the limitations of our study, the patients are from a single hospital in the Netherlands, which could introduce biases in our models given the limited patient diversity. Furthermore, the hospital houses only a few esophageal surgeons, whose postoperative complication rate depends on their skill with the procedure, which tends to improve over time. Moreover, patients are evaluated preoperatively by our surgeons as to whether they qualify for esophagectomy. Potential biases could be found through external validation with patients from other hospitals. Crucially, the esophagectomy protocols of the hospitals should be similar to ensure compatibility with our models. Besides external validation, additional data could also be used to increase the training set size. Yet, in this study we show that machine learning methods can provide a good prognosis prediction for AL, for which other studies are limited.

The relative performance of predictive models is best assessed in terms of several different metrics. The comparison of our machine learning approach versus logistic regression is summarized in Table 4. A range of metrics including the area under the curve (AUROC) as well as the recall and the F1-score are of key relevance for the complication prediction after esophagectomy. Ideally, optimal performance across all these metrics in a chosen weigthing should be pursued. However, the data available to the study presented here poses limitations to such a combined optimal performance. Rather, we illustrated the relative performance of our machine learning method versus logistic regression concentrating on the AUROC metric. This shows largely similar performance among the two approaches, underpinning the robustness with which the predictions of complications can be quantified. It does not display a definite superiority of one approach over the other as far as AUROC is concerned. Further study based on significantly larger datasets may lift some of these limitations and establish possible differences in the performance level.

Mid-fusion allows for more cross-modal information flow by concatenating hidden layer features instead of predictions like late fusion. Yet, late fusion scored higher on AUROC than mid-fusion in all cases, with differences up to 0.06. Either there were no subtle cross-modal intricacies that mid-fusion could have harnessed to improve predictions, or these intricacies require another fusion method. Transformer models are currently the state of the art in natural language processing and their architectures can be translated to multimodal classification problems [26]. These models use attention blocks to transform features from a given modality based on the features in another modality, which could enhance cross-modal information flow.

Building on improving cross-modal information flow, we experimented with joint multimodal learning (fine-tuning). Ideally, unimodal models will adjust their weights such that the multimodal prediction improves even if that weakens their unimodal prediction. However, joint multimodal learning can turn into a competition between modalities instead of cooperation, causing the model to converge to the best unimodal performance. Therefore, a multiplicative loss function was proposed, where the unimodal loss is lowered proportionally to the confidence in the other unimodal models [27].

Deep learning models have a large number of trainable parameters and depend on large datasets for good performance. However, data are often limited for specific medical procedures. Therefore, we restrained the number of trainable parameters by keeping our models shallow; see Appendix B for the number of trainable parameters per model. The dataset could be strengthened with data from other hospitals that perform the same surgical procedure, but this comes at the cost of increased heterogeneity. Furthermore, with a large enough dataset this retrospective study could simulate a prospective study, thereby strongly improving clinical relevance [26].

## 5. Conclusions

Our machine learning methods can predict post-esophagectomy AL well up to 2 days in advance, while pneumonia remains challenging. Taking into account earlier laboratory results, vital signs, and images, in other words including temporal information, slightly improved prediction performance. Preoperative patient characteristics have no predictive value regarding postoperative AL and pneumonia. Our best models for 1-day-ahead prediction of AL and pneumonia scored AUROCs of 0.87 and 0.74, respectively. We encourage future research to improve joint multimodal learning methods and consider larger datasets.

## Figures and Tables

**Figure 1 diagnostics-14-00439-f001:**
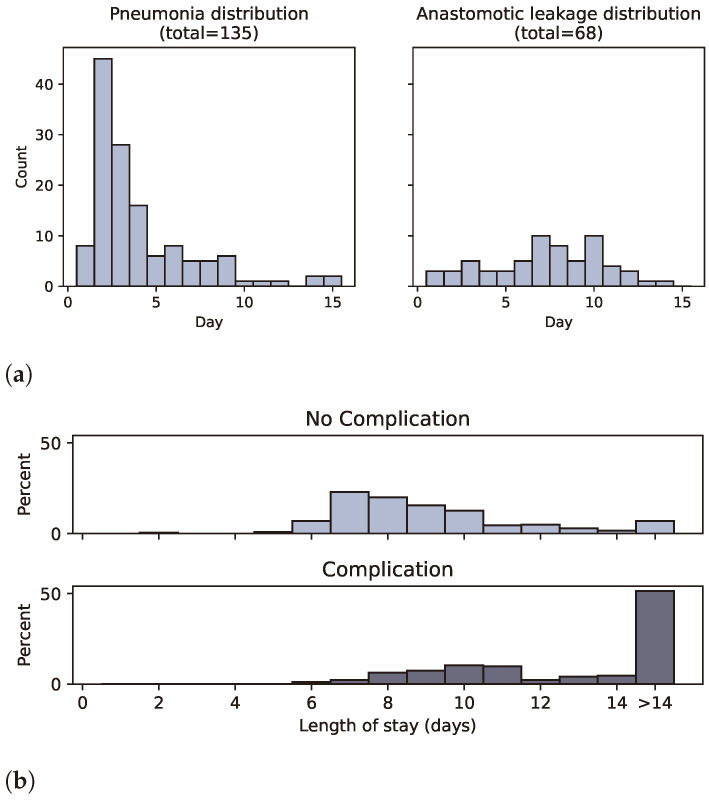
Histograms illustrating the distribution of postoperative complications. Pneumonia tends to occur early, while anastomotic leakage incidence is more evenly distributed. Complications lead to a longer hospital stay.

**Figure 2 diagnostics-14-00439-f002:**
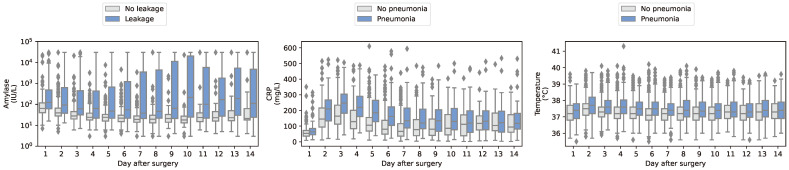
Boxplots comparing three laboratory results (rows) with three complication types (columns). Amylase values were log-10 transformed.

**Figure 3 diagnostics-14-00439-f003:**
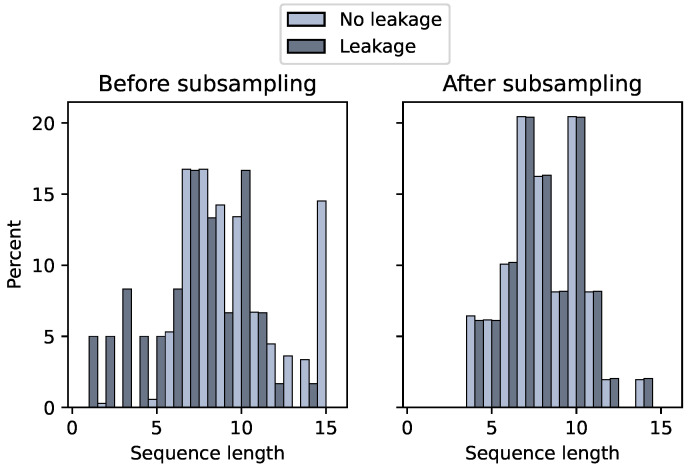
Sequence length distributions before and after data preparation.

**Figure 4 diagnostics-14-00439-f004:**
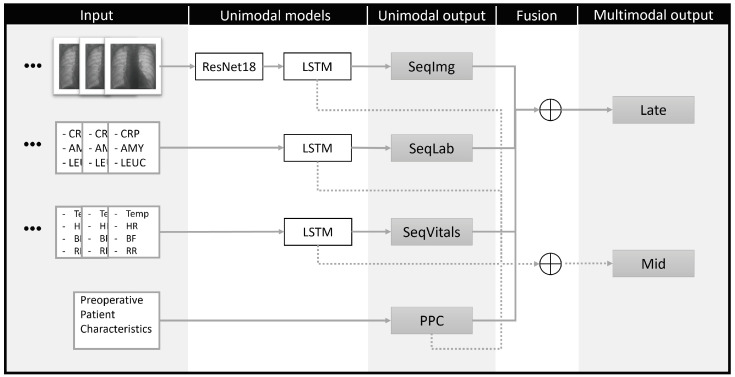
Overview of unimodal (temporal) models and how these were fused in multimodal models. Laboratory results: C-reactive protein (CRP), amylase (AMY), leucocytes (LEUC). Vital signs: temperature (Temp), heart rate (HR), respiratory rate (RR), systolic blood pressure (BP). Late fusion: Concatenation of unimodal predictions. Mid-fusion: Concatenation of unimodal hidden layer features. LSTM: long short-term memory unit.

**Table 1 diagnostics-14-00439-t001:** Summary of the preoperative patient characteristics. Includes possible value counts for binary and categorical variables. For continuous variables, the mean and standard deviation are shown.

Variable		Value (*n* = 417)
Sex	Female	85
	Male	332
Age		65.1 ± 9.0
Weight (kg)		83.0 ± 16.3
Height (cm)		176.2 ± 8.4
ASA-score	I	42
	II	240
	III	125
	IV	10
Comorbidity	Diabetes	76
	Cardiac	103
	Pulmonary	93
Surgical procedure	Ivor Lewis	329
	McKeown	88

**Table 2 diagnostics-14-00439-t002:** Clinically accepted vital sign ranges.

Vital Sign	Unit	Acceptable Range
Heart rate	bpm	[40, 200]
Temperature	°C	[35, 42]
Systolic blood pressure	mmHg	[70, 200]
Respiratory rate	breaths/min	[5, 50]

**Table 3 diagnostics-14-00439-t003:** Mean AUROC of prediction models taken from 10 × 4 and 3 × 4 cross-validation runs for LR and non-LR models, respectively. Static refers to the model with preoperative patient characteristics as input. **Bold** values indicate best test set performance for complication at given prediction window. Models with only static data have the same prediction at all windows.

Mean AUROC		Leakage						Pneumonia					
		0 h		24 h		48 h		0 h		24 h		48 h	
**Logistic Regression**		Train	Test	Train	Test	Train	Test	Train	Test	Train	Test	Train	Test
	Lab	0.92	0.92	0.84	0.83	0.76	0.75	0.76	0.74	0.68	0.66	0.61	0.57
	Vitals	0.81	0.81	0.72	0.70	0.72	0.69	0.74	0.73	0.68	0.67	0.63	0.60
	Static	0.63	0.46	-	-	-	-	0.64	0.53	-	-	-	-
	Lab+Vitals	0.92	0.92	0.82	0.81	0.78	0.76	0.77	0.74	0.74	0.71	0.65	**0.61**
	Lab+Vitals+Static	0.92	0.92	0.82	0.80	0.79	0.75	0.77	0.72	0.75	0.70	0.68	0.59
**Unimodal Models**													
	Static	0.50	0.46	-	-	-	-	0.49	0.47	-	-	-	-
	Lab	0.95	0.93	0.89	**0.87**	0.85	0.81	0.87	**0.85**	0.75	0.70	0.68	0.60
	Vitals	0.82	0.78	0.78	0.74	0.77	0.72	0.78	0.71	0.72	0.69	0.68	0.58
	Single Image	0.69	0.61	0.72	0.65	0.69	0.61	0.66	0.53	0.68	0.52	0.66	0.51
	Image Sequence	0.75	0.68	0.76	0.68	0.75	0.66	0.72	0.61	0.71	0.60	0.70	0.57
**Multimodal Models**													
	Late fusion	0.95	0.92	0.91	0.85	0.89	0.81	0.91	**0.85**	0.83	0.73	0.78	**0.61**
	Mid-fusion	0.93	0.89	0.91	0.80	0.89	0.77	0.92	0.81	0.83	0.69	0.79	0.58
	FT late fusion	0.97	**0.93**	0.93	0.84	0.91	**0.82**	0.93	0.84	0.85	**0.74**	0.78	0.59
	FT mid-fusion	0.94	0.87	0.91	0.79	0.91	0.77	0.93	0.82	0.83	0.71	0.76	0.58

**Table 4 diagnostics-14-00439-t004:** The 1-day-ahead test set prediction metrics for selected models. Mean was taken over 3 × 4 cross-validation runs.

1 Day Ahead	Leakage					Pneumonia				
	AUROC	Precision	Recall	Specificity	F1-Score	AUROC	Precision	Recall	Specificity	F1-Score
**Logistic Regression**										
Lab	0.83	0.07	0.74	0.80	0.12	0.66	0.07	0.59	0.69	0.12
Vitals	0.70	0.06	0.53	0.80	0.11	0.67	0.22	0.67	0.62	0.32
Lab+Vitals	0.81	0.07	0.69	0.78	0.12	0.71	0.09	0.70	0.69	0.13
**Temporal Models**										
Lab	0.87	0.46	0.76	0.84	0.54	0.70	0.33	0.58	0.75	0.41
Vitals	0.74	0.31	0.58	0.76	0.36	0.69	0.27	0.69	0.62	0.38
Image	0.68	0.29	0.63	0.70	0.36	0.60	0.28	0.52	0.66	0.32
Late fusion	0.85	0.40	0.72	0.84	0.51	0.73	0.33	0.65	0.72	0.42
FT late fusion	0.84	0.44	0.71	0.86	0.53	0.74	0.33	0.70	0.70	0.43

## Data Availability

The data are not publicly available due to privacy and ethical considerations. The source code, with dummy data, will be made publicly available on GitHub (https://github.com/jornjan/padeema-open-access (accessed on 1 October 2023)).

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
