# Peer review of "Complication Prediction after Esophagectomy with Machine Learning"

_diagnostics, 2024, doi:10.3390/diagnostics14040439_

Round 1
Reviewer 1 Report
Comments and Suggestions for Authors
General comments
=============
Your manuscript addresses a significant topic in the intersection of healthcare and machine learning. However, it requires substantial enhancements to clarify specific aspects, particularly the dataset nuances and the AI methodologies employed. This will ensure the content is more comprehensible and relevant to readers in the healthcare domain.
Specific comments
=============
Major comments
---------------------
1. Introduction to Complication Prediction and Machine Learning: Expand the introduction to provide a comprehensive background on the use of machine learning in predicting complications post-esophagectomy. This should include a discussion on the evolution, relevance, and current trends in this field.
2. Dataset Details: Offer a detailed description of the dataset used in your study. This should encompass the basic demographic information (age, sex), pre-operative characteristics of patients, and the indications for undergoing esophagectomy. These details are crucial for readers to understand the context and applicability of your findings.
3. Comparative Analysis of Dataset for Two Complications: Elaborate on how the dataset allows for a comparison between the two major complications studied. It's important to demonstrate that the dataset includes sufficient data to support predictions about these specific complications.
4. Rationale for Selected Laboratory Data: Justify the choice of laboratory data (CRP, WBC, amylase) used in predicting post-procedure complications. Explain why these markers are significant and perhaps why other potential markers (like serum albumin) were not considered or how they relate to patient health status.
5. Details on Multimodal Deep-Learning Model Fusion: Provide a more detailed description of how you fused multimodal deep-learning models in your study. Also, explain the rationale behind choosing this particular approach, including its benefits and how it enhances prediction accuracy.
6. Discussion of Limitations: In the discussion section, it is essential to acknowledge the limitations of your study. This could include aspects such as dataset size, diversity, potential biases, or limitations in the machine learning algorithms used.
7. Clarification on Pre-Operative Characteristics' Predictive Value: Address the conclusion that pre-operative patient characteristics do not predict postoperative AL and pneumonia. Discuss whether this is a result of dataset limitations, such as the selection of too few or inadequate characteristics.
8. Justification for AUC Scores: When concluding that pre-operative characteristics have no predictive value for postoperative AL and pneumonia, provide a rationale for why AUC scores of 0.6 or 0.7 are considered sufficient for predicting these complications. Discuss the implications of these scores in the context of machine learning efficacy and clinical relevance.
These enhancements will greatly improve the manuscript's clarity, depth, and usefulness to its intended audience in the healthcare and machine learning communities.
Comments on the Quality of English LanguagePlease check previous the quality of English Language rating.
Author Response
"Please see the attachment."

Reviewer 2 Report
Comments and Suggestions for Authors
Congratulations to the authors for their meticulous work in developing an AI-based algorithm for promptly predicting anastomotic leaks and pneumonia in the immediate postoperative period following esophagectomy.
I've included my specific comments in the pdf I've attached. file. As a general recommendation, I would suggest following the "Guidelines for Developing and Reporting Machine Learning Predictive Models in Biomedical Research" (Luo W, et al., doi: 10.2196/jmir.5870). Although these guidelines have not been endorsed by any official corpus, they offer a universally acceptable language and make the information presented in the article accessible and comprehensible even for healthcare providers who are not that familiar with AI and its applications.

Author Response
"Please see the attachment."

Reviewer 3 Report
Comments and Suggestions for Authors
Title: Predicting Anastomotic Leakage and Pneumonia after Esophagectomy Using Machine Learning: A Comprehensive Study
Comments:
Significance of the Study:
The study addresses a crucial issue in the field of esophageal cancer treatment by exploring the potential of machine learning in predicting postoperative complications such as anastomotic leakage and pneumonia. The importance of this research in improving patient outcomes is commendable.
Dataset and Features:
The utilization of a comprehensive dataset comprising 417 patients who underwent esophagectomy with multimodal temporal information, including lab results, vital signs, thorax images, and pre-operative patient characteristics, is a strength of the study. The incorporation of diverse data types enhances the robustness of the analysis.
Model Performance:
The achieved mean test set AUROC scores of 0.87 and 0.82 for predicting anastomotic leakage 1 and 2 days in advance, respectively, demonstrate the effectiveness of the machine learning models. Similarly, the AUROC scores of 0.74 and 0.61 for predicting pneumonia 1 and 2 days ahead provide valuable insights into the potential application of these models.
Clinical Implications:
The conclusion that machine learning models can effectively predict anastomotic leakage and pneumonia after esophagectomy is noteworthy. This finding could have significant implications for clinical practice, enabling proactive measures to be taken to mitigate postoperative complications and improve overall patient care.
Methodological Rigor:
The study showcases methodological rigor in handling temporal information and predicting complications in advance. The transparency in model performance metrics adds credibility to the findings.
Future Directions:
It would be beneficial to explore potential avenues for integrating these machine learning models into clinical decision support systems. Additionally, further validation studies and prospective trials could strengthen the generalizability of the findings.
Clarity of Presentation:
The paper is well-organized and clearly articulates the research question, methodology, results, and conclusions. This clarity contributes to the overall quality of the manuscript.
In summary, this paper provides a valuable contribution to the field of esophageal cancer treatment, demonstrating the potential of machine learning models in predicting postoperative complications. The thorough analysis, robust methodology, and clinically relevant findings make it suitable for acceptance.
Comments on the Quality of English Language
Title: Predicting Anastomotic Leakage and Pneumonia after Esophagectomy Using Machine Learning: A Comprehensive Study
Comments:
Significance of the Study:
The study addresses a crucial issue in the field of esophageal cancer treatment by exploring the potential of machine learning in predicting postoperative complications such as anastomotic leakage and pneumonia. The importance of this research in improving patient outcomes is commendable.
Dataset and Features:
The utilization of a comprehensive dataset comprising 417 patients who underwent esophagectomy with multimodal temporal information, including lab results, vital signs, thorax images, and pre-operative patient characteristics, is a strength of the study. The incorporation of diverse data types enhances the robustness of the analysis.
Model Performance:
The achieved mean test set AUROC scores of 0.87 and 0.82 for predicting anastomotic leakage 1 and 2 days in advance, respectively, demonstrate the effectiveness of the machine learning models. Similarly, the AUROC scores of 0.74 and 0.61 for predicting pneumonia 1 and 2 days ahead provide valuable insights into the potential application of these models.
Clinical Implications:
The conclusion that machine learning models can effectively predict anastomotic leakage and pneumonia after esophagectomy is noteworthy. This finding could have significant implications for clinical practice, enabling proactive measures to be taken to mitigate postoperative complications and improve overall patient care.
Methodological Rigor:
The study showcases methodological rigor in handling temporal information and predicting complications in advance. The transparency in model performance metrics adds credibility to the findings.
Future Directions:
It would be beneficial to explore potential avenues for integrating these machine learning models into clinical decision support systems. Additionally, further validation studies and prospective trials could strengthen the generalizability of the findings.
Clarity of Presentation:
The paper is well-organized and clearly articulates the research question, methodology, results, and conclusions. This clarity contributes to the overall quality of the manuscript.
In summary, this paper provides a valuable contribution to the field of esophageal cancer treatment, demonstrating the potential of machine learning models in predicting postoperative complications. The thorough analysis, robust methodology, and clinically relevant findings make it suitable for acceptance.
Author Response
"Please see the attachment."

Round 2
Reviewer 1 Report
Comments and Suggestions for Authors
General comments
=============
Your manuscript addresses a significant topic in the intersection of healthcare and machine learning. However, it requires substantial enhancements to clarify specific aspects, particularly the dataset nuances and the AI methodologies employed. This will ensure the content is more comprehensible and relevant to readers in the healthcare domain. Almost all responses were reasonable, except for the following.
Specific comments
=============
Major comments
---------------------
- Selected Laboratory Data: The core strength of machine learning, especially in the context of healthcare, is its ability to uncover predictive rules that may not be immediately apparent to human analysis. In this vein, the choice of parameters, particularly the Area Under the Curve (AUC), seems too constrained. I acknowledge the limitations posed by data availability, which is a common challenge in healthcare studies. However, it would be beneficial to the manuscript if these limitations were explicitly acknowledged and discussed.
=============
Minor comments
---------------------
- Correction of Patient Characteristics: There's a discrepancy in the reported number of patient characteristics. The text currently states “7 patient characteristics,” which is contradicted by the data presented in Table 1. The correct number, as per the table, is “417 patient characteristics.” This is not a trivial error, as it significantly impacts the reader's understanding of the scope and depth of the study. I recommend a revision of this sentence to accurately reflect the data presented.
In conclusion, the manuscript has the potential to contribute valuably to the literature on healthcare and machine learning. The suggested revisions are intended to refine and clarify the study, thereby enhancing its impact and utility for the intended audience.
Author Response
"Please see the attachment."
